# Effectiveness of Xpert MTB/RIF and the Line Probe Assay tests for the rapid detection of drug-resistant tuberculosis in the Central African Republic

**Alain Farra**[1]*, **Karen Koula**[2‡], **Boris Lokoti Jolly**[1‡], **Hervé Gildas Gando**[3‡], **Louis Médart Ouarandji**[3‡], **Christian Diamant Mossoro-Kpinde**[2‡], **Alexandre Manirakiza**[4‡], **Jean Pierre Simelo**[5], **Jean de Dieu Iragena**[6]

1 National Reference Laboratory for Tuberculosis, Institut Pasteur of Bangui, Bangui, Central African Republic, 2 Department of Microbiology, University of Bangui, Bangui, Central African Republic, 3 Coordination Unit of the National Tuberculosis Control Program, Ministry of Health, Bangui, Central African Republic, 4 Epidémiology Unit, Institut Pasteur of Bangui, Bangui, Central African Republic, 5 TB/HIV and Laboratories, Kinshasa, Democratic Republic of the Congo, 6 HIV/TB and Sexually Transmetted Infections, WHO/AFRO, Brazzaville, Congo

☯ These authors contributed equally to this work.
‡ KK, BLJ, HGG, LMO, CDMK and AM also contributed equally to this work.
* farra_alain@yahoo.fr, alain.farra@pasteur-bangui.cf

**Data Availability Statement:** The data is attached as supporting information.

## Abstract

The Xpert MTB/RIF and Line Probe Assay (LPA) tests are more and more frequently used in mycobacteria testing laboratories for the rapid diagnosis of multi-drug resistance (MDR-TB). In this study, we demonstrate the effectiveness of these tests in the Central African Republic. Rifampicin resistance cases detected by the Xpert MTB/RIF during the year 2020 are also underwent first- and second-line LPA, and a first-line of drug susceptibility testing (DST) on solid medium and we compared these results. 101 rifampicin resistance cases based on the Xpert MTB/RIF were detected. Mean age was 34 years [16–81]. The 20–40 years age group represented 73.2% and the male-to-female sex ratio was 1.9:1. Patient profiles were dominated by treatment failure cases (40.6%) followed by relapsed cases (30.7%) and new cases (18.8%). These 101 rifampicin resistance were also detected with the first-line LPA and were confirmed by the DST. Similarly, the isoniazid results obtained with the first-line LPA, were confirmed by the DST, giving a concordance of 100% for these antibiotics. Rifampicin resistance were for the most part due to the absence of the WT8 sequence (56%) and the presence of the Mut3 mutation (53.4%). The majority of the isoniazid resistance (94.2%) were due to the Mut1 mutation in the *katG* gene and 4.2% of the cases involved both the *katG* gene and the *inhA* gene promoter with the Mut1 mutation. The second-line LPA test no resistance to second-line antibiotics. This study demonstrated the effectiveness of the Xpert MTB/RIF and the LPA tests for the rapid diagnosis of MDR-TB in the Central African Republic. However, due to their high cost, these tests have not been extensively deployed in the country. Public authorities and their TB-partners can help make these molecular tests more accessible to fight MDR-TB in the country.

**Funding:** The authors received no specific funding for this work.

**Competing interests:** The authors declare that they have no competing interest.

## Introduction

The emergence and the spread of multidrug-resistant tuberculosis (MDR-TB) not only constitute a real public health issue, but also represent a real challenge for the World Health Organization (WHO)'s "End TB Strategy" that aims to virtually eliminate TB by 2035. The use of molecular tests such as the Xpert MTB/RIF and Line Probe Assay (LPA) tests have drastically enhanced the treatment of MDR-TB by providing rapid diagnosis and thus detection of resistance to anti-TB drugs. Conventional drug-susceptibility tests (DST), based on cultures and susceptibility tests on solid Lowënstein-Jensen (LJ) agar, are time-consuming and require long turnaround times of up to 3 months. For this reason, the WHO now recommends molecular tests carried out directly on sputum samples, e.g. the Xpert MTB/RIF test for TB-positive or -negative sputum samples and LPA for positive sputum samples or on culture samples [1–3]. Regarding rifampicin, these molecular techniques amplify the rifampicin resistance-determining region (RRDR) of the *rpoB* gene, a short sequence of 81 bp including the main mutations that confer resistance to the rifampicin antibiotic. These mutations occur primarily in codons 526 and 531 and secondarily in codons 511, 516, 518, 522 and 533 [4, 5]. In addition to rifampicin resistance, the first-line LPA test also detects isoniazid resistance based on mutations in the *katG* gene (codon 315) and those in the promoter of the *inhA* gene involving codons 8, 15 and 16 [5]. The second-line LPA test, indicated for MDR-TB or rifampicin-resistant TB (RR-TB) cases, detects resistance to second-line drugs, namely to fluoroquinolones (FLQ) and injectable drugs such as amikacin (AMK), kanamycin (KAN) and capreomycin (CAP). The second-line LPA tests for FLQ resistance from mutations affecting codons 90, 91 and 94 of the *gyrA* gene. Resistance to injectable anti-TB drugs involves mutations in the *rrs* gene on codons 1401 and 1402, associated with resistance to AMK and KAN, and mutations on codon 1484, associated with resistance to all three drugs (AMK, KAN and CAP) [6].

The Central African Republic (CAR) is one of the 30 countries in the world with a high tuberculosis burden, with more than 500 cases for 100,000 population [7]. Surveillance of MDR-TB on the national level is not extensive, given the very limited deployment of GeneXpert machines. Cultures and phenotypic susceptibility tests can only be carried out at the National Reference Laboratory for Tuberculosis (NRL-TB) at the Institut Pasteur of Bangui, due to the lack of adequate testing facilities in the country. The national TB control program (NTCP) began to establish the use of the Xpert MTB/RIF test in early 2020 as the first-line test at the NRL-TB. The NTCP adopted this method following WHO recommendations, because molecular tests allow for rapid diagnosis of TB and the detection of TB drug resistance through the national TB surveillance plan. Here, we set out to demonstrate the effectiveness of the Xpert MTB/RIF and LPA tests as efficient means for the rapid detection of MDR-TB in the CAR.

## Materials and methods

### Study type and duration

We carried out a cross-sectional study for one year, from 1 January to 31 December 2020, on patients in whom RR-TB was detected via the Xpert MTB/RIF test and documented at the NRL-TB at the Institut Pasteur of Bangui.

### Study population

The study population was made up of patients putatively affected by MDR-TB as targeted by the NTCP, namely treatment failures, relapses, lost-to-follow-up and MDR-TB contact cases as well as presumptive TB cases. The patients had been referred to the NRL-TB by local

community diagnosis and treatment centers (DTCs). Patients brought their sputum with the NTP examination request form filled out at the DTC for molecular, culture and drug succeptibility tests (DST); minor patients were accompanied by their parents or guardians. Some sputum samples from patients in outlying provinces were also sent with the NTP test request form to the NRL-TB by healthcare personnel working with aid organizations. These samples arrived within 48 h in UN 2814-certified triple-packaging transportation boxes.

**Inclusion criteria.** All patients for whom the Xpert MTB/RIF test detected rifampicin resistance were included in this study.

**Exclusion criteria.** The patients for whom the Xpert MTB/RIF test results were negative, invalid or undetermined after two repetitions or positive without detection of rifampicin resistance, were excluded from this study.

## Laboratory analyses

Laboratory analyses were done according to workflow for MDR-TB defined by the NTP and used at the NRL-TB (Fig 1).

**The Xpert MTB/RIF test.** The Xpert MTB/RIF was carried out on all sputum samples using a GeneXpert 4 module instrument (version 4.7). To carry out this test, one volume of the sputum sample was homogenized with 2 volumes of decontamination and lysis solution of the kit and incubated for 15 min. After this incubation time, 2 mL of the clear solution was pipetted and deposited in the cartridge, and then placed in the GeneXpert 4 instrument. Results were obtained in less than 2 hours thereafter. According to the current NRL-TB algorithm, when the Xpert MTB/RIF detects rifampicin resistance, the LPA tests were performed

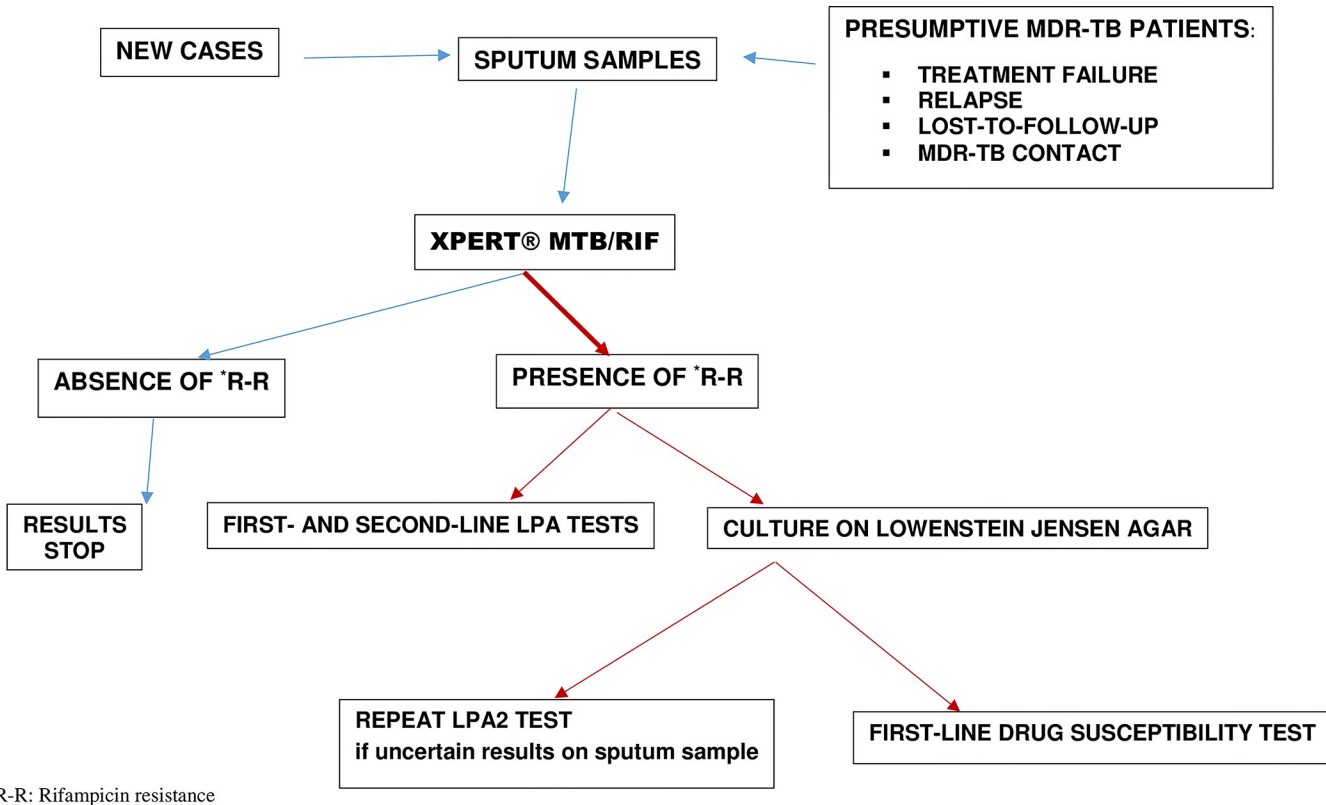

*R-R: Rifampicin resistance

**Fig 1. Workflow for the diagnosis of MDR-TB on sputum samples.**

as well as the culture and DST. If rifampicin resistance is not detected, or if the results are indeterminate or invalid, the LPA, culture and DST are not performed and the results are returned to the patient.

**The GenoType MTBDR (ver. 2) assay or the Line Probe Assay (LPA).** These tests were carried out on sputum samples for which the Xpert MTB/RIF test had detected rifampicin resistance and on some culture samples. These tests were performed in three steps in dedicated facilities, i.e. DNA extraction, resistance gene amplification and hybridization. DNA was extracted using the alkaline lysis method in the GenoLyse kit provided with the reagent. PCR amplification was run in a GTQ-Cycler 96 (Hain Lifescience) thermocycler and hybridization was carried out in a TwinCubator (Hain Lifescience) with nitrocellulose DNA strips on which the complementary probes of the main wild-type (non-mutant) and mutant sequences involved in MDR-TB have been bound.

**GenoType MTBDRplus (first-line LPA).** This test was carried out on all of the sputum samples, which had been decontaminated using the Petroff method with 4% NaOH prior to the LPA test. This first-line test detects the *Mycobacterium* complex, rifampicin resistance involving the main mutations of the *rpoB* gene and isoniazid resistance involving the *katG* gene and the *inhA* gene promoter (Fig 2). The identification of these resistance genes made it possible to compare the results from the Xpert MTB/RIF test with the first-line LPA test and the first-line DST regarding rifampicin and isoniazid, the two major first-line antibiotic treatments (Fig 2).

**GenoType MTBDRsl (second-line LPA).** We also tested sputum samples and some culture samples following ambiguous test results on the DNA strips. In these cases, extraction was performed directly on culture samples without a decontamination step. The second-line LPA test is designed to identify the *Mycobacterium* complex, FLQ resistance (e.g. ofloxacin (OFX) and moxifloxacin (MFX)) by detecting mutations in the *gyrA* and *gyrB* genes, resistance to injectable antibiotics used in second-line treatment such as KAN, AMK and CAP by detecting mutations in the *rrs* gene and, finally, mutations of *eis* gene for detection of a low-level KAN-resistance.

## Culture and drug susceptibility tests

They were performed in the level 2+ laboratory of the NRL-TB operating with a negative pressure system with dedicated Type II Microbiological Safety Stations for culture and sensitivity testing. Cultures were carried out on solid LJ agar with two tubes per patient incubated at 37°C for up to 2 months. Cultures were followed by DST on LJ agar using the proportion method for first-line TB drugs, namely rifampicin, isoniazid and ethambutol. Susceptibility testing for second-line TB drugs is not carried out at the NRL-TB in Bangui.

Although the rate of staining has always been less than 3% in our laboratory, the rare cases when they occur are systematically repeated in order to reduce the waiting time for results.

## Data collection and analysis

The data collected from the examination request forms and the results from the different analyses were recorded in a spreadsheet. Sampling was exhaustive, including all patients declared positive with the Xpert MTB/RIF test during the year 2020. The socio-demographic characteristics of the patients as well as results on drug resistance were analyzed using Stata software (version 14).

## Ethical considerations

This MDR-TB surveillance program was approved by the ethics committee of the Ministry of Health as part of the NTCP in the CAR (RCA PSN TB 2017–2023). The request for

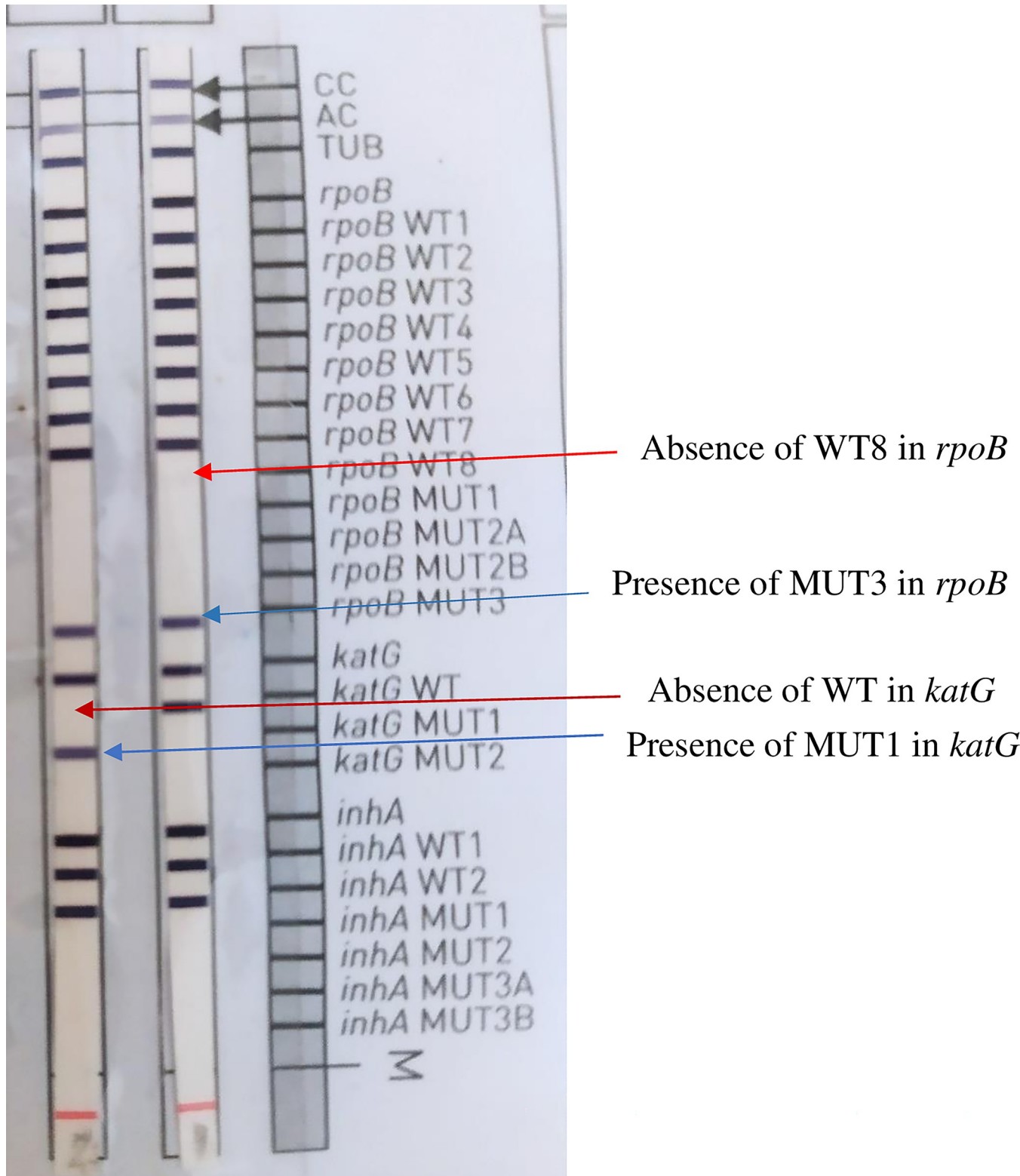

**Fig 2. Results of DNA hybridization in the first-line LPA, showing a mutation in the *rpoB* gene and in the *katG* gene.**

examination forms are the ones designed by the NTP and filled out at the DTC. It was therefore not necessary for this study to establish informed consent requests for patients of any age at the NRL-TB. However, the data analysis was done anonymously and with strict respect for the confidentiality and identity of the patients included. All results were systematically transmitted to the NTP, which organizes the management and follow-up of MDR-TB patients at the MDR-TB treatment center in Bangui. The dissemination of the results of this study is in line with the operational research policy on tuberculosis defined and approved by the Ministry of Health.

## Results

In total, 1404 patients were registered for the Xpert MTB/RIF test of whom 458 (32.6%) tested positive for TB. Among these TB-positive patients, 101 (22%) showed RR-TB and thus made up our study population.

### Characteristics of the study population

The average age was 34 years, ranging from 16 to 81 years. The 21–40 age group represented nearly three-quarters of the MDR-TB patients, accounting for 73.2% (74/101) of the cases. Males predominated this population with 66.3% (67/101) of the cases; the sex ratio was 1.9:1. The vast majority (95.1%) of patients were from the capital Bangui (96/101). Treatment failure and relapse cases made up the highest proportions of resistance cases, with respectively 40.6% (41/101) and 30.7% (31/101) of the cases; new cases represented 18.8% (19/101) of the study population. Regarding the resistance profile, 85.2% (86/101) showed both rifampicin and isoniazid resistance compared with 14.8% (15/101) showing resistance only to rifampicin (Table 1).

### Concordance of results for rifampicin and isoniazid resistance

All samples showing rifampicin resistance with the Xpert MTB/RIF test were also determined resistant with the first-line LPA test as well as with the first-line DST, with 100% concordance between all three detection methods.

**Table 1. Socio-demographic characteristics and resistance profile of the study population.**

| Characteristics | N (%) |
| --- | --- |
| *Age group (years)* | |
| ≤ 20 | 5 (5%) |
| 21–40 | 74 (73.2%) |
| 41–60 | 20 (19.8%) |
| ≥ 61 | 2 (2%) |
| *Gender* | |
| Male | 67 (66.3%) |
| Female | 34 (33.7%) |
| *Origin* | |
| Bangui | 96 (95.1%) |
| Provinces | 5 (4.9%) |
| *Types of cases* | |
| New cases | 19 (18.8%) |
| Lost-to-follow-up | 8 (7.9%) |
| Relapse | 31 (30.7%) |
| Treatment failure | 41 (40.6%) |
| MDR-TB contact | 2 (2%) |

**Table 2. Results of the Xpert MTB/RIF, first-line LPA and first-line DST tests.**

| Resistance profile | Xpert MTB/RIF | First-line LPA (%) | First-line DST (%) |
|---|---|---|---|
| Rifampicin resistant | 101 | 101 (100) | 101 (100) |
| Rifampicin susceptible | - | | 0 |
| Isoniazid resistant | *NA | 86 (85.2) | 86 (100) |
| Isoniazid susceptible | *NA | 15 (14.8) | 15 (100) |

*NA: Not applicable

Likewise, for isoniazid resistance, resistance and drug susceptibility were also determined with the first-line LPA and the first-line DST, again with 100% concordance (Table 2).

***rpoB* gene wild-type and mutation probes observed in the first-line LPA.** The WT8 probe was absent in 56.6% (57/101) of the samples and these absences were all associated with Mut3 mutations, accounting for 53.4% (54/101) of the resistance cases. The WT7 probe was absent in 21.9% of the resistant samples, and this profile was associated with the Mut2 mutation in 11.9% (12/101) and Mut2B in 5.9% (6/101) of the cases.

The absence of hybridization for two or more WT sequences was observed, with the absence WT2 –WT3, WT3 –WT4 or WT5 –WT6 and a triple absence WT3 –WT4 –WT7; these profiles were observed in variable proportions in association with the Mut1 mutation. In 13.9% (14/101) of the cases, the absence of WT probe hybridization was not accompanied by hybridization with a mutation probe. In addition, no mutation probes developed in the presence of a positive WT probe (Table 3).

***katG* gene wild-type and mutation probes observed in the first-line LPA.** Of the 86 cases of isoniazid resistance, 81 (94.2%) cases with an absent WT probe were observed in the *katG* gene; all developed a Mut1 band. No Mut2 bands were observed and no mutation bands developed in the presence of a WT probe band (Table 4).

***inhA* gene wild-type and mutation probes observed in the first-line LPA.** For the *inhA* gene, the study showed 4 WT1 absences, 3 WT2 absences and 2 WT1 –WT2 double absences, all associated with a Mut1 band. No Mut2, Mut3A or Mut3B probe bands were observed and no WT absences were observed without a defined mutation probe band (Table 5). In addition, four samples developed both an absence of WT probe and a mutation for the *katG* and *inhA* genes.

## Results of the second-line LPA test

The analyses carried out with the second-line LPA test did not reveal any WT absences or any mutations in the *gyrA* or *gyrB* genes that confer resistance to FLQ. Likewise, no WT absences

**Table 3. First-line LPA results, with hybridization to wild-type (WT) and mutation probe sequences for the *rpoB* gene.**

| WT probe(s) | Absence of WT (%) | Absence of mutation | Mut1 | Mut2A | Mut2B | Mut3 |
|---|---|---|---|---|---|---|
| WT2 | 1 (0.9) | 1 | - | - | - | - |
| WT2 –WT3 | 2 (1.9) | 2 | - | - | - | - |
| WT3 –WT4 | 14 (13.9) | 4 | 10 | - | - | - |
| WT3 –WT4 –WT7 | 2 (1.9) | - | 2 | - | - | - |
| WT5 –WT6 | 3 (2.9) | - | 3 | - | - | - |
| WT7 | 22 (21.9) | 4 | - | 12 | 6 | - |
| WT8 | 57 (56.6) | 3 | - | - | - | 54 |
| **TOTAL** | **101** | **14** | **15** | **12** | **6** | **54** |
| | | **(13.9%)** | **(14.8%)** | **(11.9%)** | **(5.9%)** | **(53.4%)** |

**Table 4. Wild-type and mutation profiles observed for the *katG* gene in patients showing isoniazid resistance with the first-line LPA test.**

| Absence of WT | Absence of mutation | MUT1 | MUT2 |
|---|---|---|---|
| 81 | - | 81 | - |

**Table 5. Wild-type and mutation profiles observed for the *inhA* gene in patients showing isoniazid resistance with the first-line LPA test.**

| WT | Absence of WT | Absence of mutation | Mut1 | Mut2 | Mut3A | Mut3B |
|---|---|---|---|---|---|---|
| WT1 | 4 | - | 4 | - | - | - |
| WT2 | 3 | - | 3 | - | - | - |
| WT1 –WT2 | 2 | - | 2 | - | - | - |

or mutations were recorded for the *rrs* or *eis* gene that confers resistance to KAN, AMK and CAP. The second line LPA did not show any case of pre-XDR or XDR.

## Discussion

This study set out to demonstrate the effectiveness of molecular tests for the rapid diagnosis of drug-resistant tuberculosis. These tests were carried out according to the current procedures used at the NRL-TB in the CAR, which regularly undergoes external review for quality control by the supranational reference laboratory (SRL) for TB in Cotonou, Benin with regard to the Xpert MTB/RIF test and by the SRL for TB in Antwerp, Belgium with regard to first-line DST and LPA. The limitation of this study could be the absence of sequencing for the 14 rifampicin resistance cases where the absence of WT on the *rpoB* had not shown visible mutations after hybridization on the strips. But according to the Global Laboratory Initiative guideline, the absence of WT without the appearance of a mutation defines a resistance that sequencing will make it possible to specify [2]. Also, our phenotypic (DST) results helped to support these conclusions. In any case, as reported in other studies, future studies should include sequencing to identify the specific mutations underlying the absence of a WT probe [4, 8, 9].

Patients with RR-TB or MDR-TB were primarily male (66.3%, 67/101) and young (aged between 20 and 40 years; 73%, 74/101). That young men show higher incidence of MDR-TB may be linked to their socio-professional situation that increases their exposure to TB in general and thus MDR-TB [10]. Treatment failure and relapse patients accounted for the majority of the MDR-TB cases, representing 40.6% (41/101) and 30.7% (31/101) of the cases, respectively. In contrast to lost-to-follow-up cases, treatment failure and relapse cases are known for their long exposure to TB drugs, making them particularly prone to MDR-TB [11, 12]. The proportion of new cases among the MDR-TB cases was 18.8% (19/101); this rather high figure is of great concern, perhaps reflecting the difficulties the NTCP encounters to control MDR-TB in the CAR. However, relative to new TB cases, the MDR-TB rate is comparatively low. Nonetheless, the NTCP must intensify its surveillance efforts, because MDR-TB can spread if adequate TB control measures are not taken [13].

In this study, there was no discrepancy between the Xpert MTB/RIF test and the first-line LPA test regarding rifampicin resistance. Likewise, the first-line DST test confirmed the results on rifampicin and isoniazid resistance. Similar results have been reported in South Africa and in Georgia, with 100% concordance between the Xpert MTB/RIF and first-line LPA tests [14, 15]. However, discrepancies between Xpert MTB/RIF and first-line LPA tests have been reported in some studies. These discrepancies may be due to the methods used, because first-line LPA tests are sometimes carried out on sputum samples for which the Xpert MTB/RIF

test results have been interpreted as "invalid" or "error". Some discrepancies are also due to the interpretation of LPA results. Nevertheless, the conclusions of these studies generally concur that despite some discrepancies, both tests are effective for the rapid diagnosis of MDR-TB [16–20].

The DNA hybridization tests indicated a predominant absence of the WT8 sequence for the *rpoB* gene (56%), frequently in association with the Mut3 mutation (53.4%). These results echo those of many studies that also reveal an association of the absence of WT8 and the presence of the Mut3 mutation, with some reaching up to 80% of isolates. The absence of WT8 corresponds to deletions in codon 531 leading to the Ser531Leu mutation; codon 531 appears to be more commonly associated with mutations that cause rifampicin resistance [11, 21, 22].

For the *katG* gene, the Mut1, i.e. Ser315Thr, mutation was observed in association with the absence of the WT sequence. This mutation is the most frequently reported mutation for this gene, resulting from an amino acid substitution on codon 315. For the *inhA* gene, the only observed mutation was Mut1, i.e. Cys15Thr, generally associated with an amino acid substitution on codon 15 of the gene [23].

Regarding the second-line LPA tests, our study did not reveal any mutations in genes conferring resistance to second-line drugs. However, it is important to continue to monitor resistance to second-line drugs, not only with the LPA test, but also with Xpert MTB/XDR tests and in liquid cultures, which allow testing on a wider range of second-line drugs, including bedaquiline.

## Conclusion

This study demonstrated the effectiveness of the Xpert MTB/RIF and LPA tests for the rapid diagnosis of MDR-TB. The cases of resistance revealed using these molecular tests were all confirmed with the phenotypic DST test. The use of Xpert MTB/RIF as a first-line test has allowed the identification of MDR-TB in new patients who may have mistakenly benefited from first-line treatment and not only progressed to treatment failure but also spread MDR-TB in the community. In addition to saving considerable time, these tests are an important tool for both the diagnosis of tuberculosis and for the surveillance of MDR-TB on a national scale.

However, the use of these tests is limited, particularly in the CAR, due to their relatively high cost. Therefore, public authorities and their partners in the fight against TB, such as the WHO, the International Union Against Tuberculosis and Lung Disease and the Global Fund, must work together to make these diagnosis methods widely available to ensure the effective control of TB and particularly MDR-TB in the CAR.

## Supporting information

**S1 Data. Xpert MTBRIF, LPA and DST anonymized laboratory data base.**
(XLSX)

**S1 Text. NTP request form for microscopy, molecular test, culture and antibiogram.**
(PDF)

## Acknowledgments

We thank the personnel of the diagnosis and treatment centers that referred patients as soon as they were suspected of having MDR-TB.

We thank the Global Fund to Fight AIDS, Tuberculosis and Malaria for its support of the NTRL, which made it possible to obtain the reagents to perform the free Xpert MTB/RIF and

LPA tests. Finally, we thank the patients who agreed to undergo these examinations and who thus allowed the NTP to obtain data on TB-MR in the country.

## Author Contributions

**Conceptualization:** Alain Farra.

**Data curation:** Alain Farra, Karen Koula, Boris Lokoti Jolly, Alexandre Manirakiza.

**Formal analysis:** Alexandre Manirakiza.

**Funding acquisition:** Hervé Gildas Gando, Louis Médart Ouarandji.

**Investigation:** Alain Farra, Karen Koula, Boris Lokoti Jolly.

**Methodology:** Alain Farra, Christian Diamant Mossoro-Kpinde.

**Project administration:** Alain Farra, Hervé Gildas Gando.

**Resources:** Hervé Gildas Gando, Louis Médart Ouarandji, Jean Pierre Simelo, Jean de Dieu Iragena.

**Software:** Alexandre Manirakiza.

**Supervision:** Boris Lokoti Jolly, Christian Diamant Mossoro-Kpinde.

**Validation:** Boris Lokoti Jolly, Christian Diamant Mossoro-Kpinde, Alexandre Manirakiza, Jean Pierre Simelo, Jean de Dieu Iragena.

**Visualization:** Karen Koula, Hervé Gildas Gando, Louis Médart Ouarandji, Christian Diamant Mossoro-Kpinde, Jean Pierre Simelo, Jean de Dieu Iragena.

**Writing – original draft:** Alain Farra.

**Writing – review & editing:** Alain Farra, Christian Diamant Mossoro-Kpinde, Alexandre Manirakiza, Jean Pierre Simelo, Jean de Dieu Iragena.

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
