## [Decision Letter · Decision Letter 0]

20 Mar 2023

PGPH-D-23-00214

Effectiveness of Xpert® MTB/RIF and the Line Probe Assay tests for the rapid detection of drug-resistant tuberculosis in the Central African Republic

Dear Dr. Farra,

Thank you for submitting your manuscript to PLOS Global Public Health. After careful consideration, we feel that it has merit but does not fully meet PLOS Global Public Health’s publication criteria as it currently stands. Therefore, we invite you to submit a revised version of the manuscript that addresses the points raised during the review process.

EDITOR: Please insert comments here and delete this placeholder text when finished. Be sure to:

Indicate which changes you require for acceptance versus which changes you recommendAddress any conflicts between the reviews so that it's clear which advice the authors should followProvide specific feedback from your evaluation of the manuscript

Please ensure that your decision is justified on PLOS Global Public Health’s publication criteria and not, for example, on novelty or perceived impact.

We look forward to receiving your revised manuscript.

Kind regards,

Sara Suliman

Academic Editor

Journal Requirements:

1.  Please indicate by return email the full and correct funding information for your study and confirm the order in which funding contributions should appear.

Additional Editor Comments (if provided):

Reviewers' comments:

Reviewer's Responses to Questions

**Comments to the Author**

1. Does this manuscript meet PLOS Global Public Health’s publication criteria? Is the manuscript technically sound, and do the data support the conclusions? The manuscript must describe methodologically and ethically rigorous research with conclusions that are appropriately drawn based on the data presented.

Reviewer #1: Yes

Reviewer #2: Yes

2. Has the statistical analysis been performed appropriately and rigorously?

Reviewer #1: Yes

Reviewer #2: Yes

3. Have the authors made all data underlying the findings in their manuscript fully available (please refer to the Data Availability Statement at the start of the manuscript PDF file)?

Reviewer #1: Yes

Reviewer #2: Yes

4. Is the manuscript presented in an intelligible fashion and written in standard English?

Reviewer #1: Yes

Reviewer #2: Yes

5. Review Comments to the Author

Reviewer #1: In the manuscript, the authors described the effectiveness of Xpert MTB/RIF and LPA to detect rifampin and INH resistance in the Central African Republic. In this setting, MDR testing is centralized, and so understanding the effectiveness of these assays could support broader implementation and adoption in the country, and is of important public health value.

COMMENTS

1. If a sample had an invalid Xpert result, how was this handled by the lab workflow, was it repeated, or culture done? Please clarify in the methods and any limitations

2. A limitation in understanding the effectiveness of Xpert MTB/RIF is that culture/LPA is not done in those that are RIF negative to confirm the finding or determine if it is a false negative, and should be noted in the limitations

3. Similarly, INH resistance is only assessed in those with positive RIF, and so this will be an ongoing limitation in first-line implementation of Xpert MTB/RIF to assess for INH mono-resistance, and the authors could add this to the discussion on the implementation of MDR/XDR molecular TB testing.

4. On Line 193-194, the authors noted “all absences” in WT8 were associated with mutation 3, but 3 had no mutation noted (Table 3). Please clarify

5. Why is there an inhA table but no katg table? Why not have a combined table with 2 sections?

6. In the discussion, would highlight majority of RIF resistant cases were also INH resistant, highlighting importance to consider empiric MDR treatment if Rif resistant

7. Was there access to the rpoB probes and mutations that were detected by Xpert to compare to LPA?

8. To confirm, all Xpert MTB/RIF positive with +RIF cases were also culture positive without any contamination? If so, please indicate that in the text. Or were RIF positive cases with positive cultures selected for this analysis? Please clarify

9. As a minor comment, the focus is on RIF and INH resistance, and so the presentation of the 2nd line testing is unclear. As none were negative, would consider to just have in the results that all cases had second line testing and were MDR and none pre-xdr or xdr, and remove Figure 3

Reviewer #2: This paper describes a straightforward study assessing the concordance of GeneXpert testing for Rif resistance with first and second line LPA and culture DST. The cross-sectional study design is appropriate to the question. The authors conclusion of strong concordance between Xpert and the other DST diagnostics is supported by the data.

Major:

- in the introduction the authors should justify why it is important to asses the performance of Xpert Rif detection specifically in the CAR since Xpert already has WHO approval.

- Similarly I'd like the authors to add in the discussion section what will or should change because of these results. The performance of Xpert is well documented, and unfortunately we have not been able to meaningfully lower the price of the test. So are these results useful to lobby the NRL to spend more on these tests? To confirm that they should continue to be used in the CAR? What's the next step

Minor:

-The included figures are showing up as blurry. I would suggest uploading a higher resolution for publication

6. PLOS authors have the option to publish the peer review history of their article (what does this mean?). If published, this will include your full peer review and any attached files.

**Do you want your identity to be public for this peer review?** For information about this choice, including consent withdrawal, please see our Privacy Policy.

Reviewer #1: No

Reviewer #2: No

---

## [Editor Report · Decision Letter 1]

4 Apr 2023

Effectiveness of Xpert® MTB/RIF and the Line Probe Assay tests for the rapid detection of drug-resistant tuberculosis in the Central African Republic

PGPH-D-23-00214R1

Dear Dr. Farra,

We are pleased to inform you that your manuscript 'Effectiveness of Xpert® MTB/RIF and the Line Probe Assay tests for the rapid detection of drug-resistant tuberculosis in the Central African Republic' has been provisionally accepted for publication in PLOS Global Public Health.

Best regards,

Sara Suliman

Academic Editor

The authors have addressed the reviewers' comments.